# Investigation of the Constitutive Model of W/PMMA Composite Microcellular Foams

**DOI:** 10.3390/polym11071136

**Published:** 2019-07-03

**Authors:** Yuxuan Zhu, Guoqiang Luo, Ruizhi Zhang, Qiwen Liu, Yi Sun, Jian Zhang, Qiang Shen, Lianmeng Zhang

**Affiliations:** 1State Key Lab of Advanced Technology for Materials Synthesis and Processing, Wuhan University of Technology, 122# Luoshi Road, Wuhan 430070, China; 2Hubei Key Lab of Theory and Application of Advanced Materials Mechanics, Department of Mechanics and Engineering Structure, Wuhan University of Technology, 122# Luoshi Road, Wuhan 430070, China

**Keywords:** composite material, microcellular foam, constitutive model, data fitting, parameter identification

## Abstract

Investigating the constitutive relationship of a material can provide better understanding of the mechanical properties of a material and has an irreplaceable effect on optimizing the performance of a material. This paper investigated a constitutive model for tungsten/polymethyl-methacrylate (W/PMMA) composite microcellular foams prepared by using melt mixing and supercritical carbon dioxide foaming. The stress-strain relationships of these foams with different W contents were measured under static compression. The elastic modulus and compressive strength values of the foams were remarkably greater than those of the pure PMMA foams: at a W content of 20 wt %, these values were increased by 269.1% and 123.9%, respectively. Based on the Maxwell constitutive model, the relevant coefficients were fitted according to the experimental data of different relative densities and W contents in quasi-static compression. According to the numerical relationships between the relevant coefficients and the relative densities and W contents, the quasi-static mechanical constitutive model of W/PMMA composite microcellular foams with W contents of 0~60 wt % and relative densities of 0.15~0.55 were predicted. This study provided basic data for the optimal design of the W/PMMA composite microcellular foams and proposed a method for investigating the mechanical properties of composite microcellular foam materials.

## 1. Introduction

The constitutive relationship is the relationship between the stress tensor and the strain tensor, which is a comprehensive reflection of the macroscopic mechanical properties of a structure or material. Investigating the constitutive relationship can provide better understanding of the mechanical properties of a material and has an irreplaceable effect on the performance optimization of a material [1,2,3,4,5,6,7]. Foam materials are widely used in automotive, helmet, aerospace, and transportation packaging fields because of their light weight, flexible design, good cushioning and shock absorption properties, etc. [8,9,10,11,12]. To reasonably design a buffer structure and avoid resource waste and economic loss, it is necessary to accurately grasp the constitutive relationship of a buffer material or structure. The design of structural elements for impact safety and crashworthiness analysis is important [13,14,15,16,17,18], and usually based on finite element methods. The constitutive relationships of these materials must be fully understood in order to provide material parameter input into the finite element model [19,20,21,22,23,24].

Many researchers have proposed simplified constitutive equations that can be input into finite element models for impact simulation calculations. Rusch et al. [25] proposed a simple and general power exponential model, which can only be used to simply fit experimental data, i.e., the model cannot explain the effect of density. The Gibson-Ashby model, which is based on structural behavior [26,27], can explain the effect of density, but the microstructure of polymer foam needs to be analyzed, and the applicability of the model is limited. Both Subhash [28] and Avalle [29] proposed phenomenological constitutive models suitable for structural foams subjected to large deformations; these models can fully represent the three characteristics of compressive stress-strain response, i.e., linear elasticity, plastic plateau and densification stage. Although these models can systematically change the model parameters to account for the effect of density, it was not possible to quantitatively analyze the effect of foam density on the model parameters. Voga [30] proposed a five-parameter model named the Maxwell model. The Maxwell model can be used to predict the stress-strain curve of a foam and obtain the relationship between the parameters and the relative density. The use of the Maxwell model can reduce the number of experimental tests. Each parameter in the model has a mechanical significance, which can help researchers to better study the mechanical properties of foam materials.

In this study, we prepared W/PMMA composite microcellular foams by melt-mixing and supercritical carbon dioxide foaming methods and then tested the static compression mechanical properties of these composite microcellular foam materials. Studies have shown that density is the most important feature affecting foam compression performance. [26] We used the Maxwell model to fit the static compressive stress-strain curves of the W/PMMA composite microcellular foam materials. The experimental data accurately reflected the typical behavior of these microcellular foams and were consistent with the model calculation results, which verified the accuracy of the model. Through a series of model parameter identifications of different relative densities and W contents, we innovatively obtained the relationship between relative density, W content and model parameters in the static mechanical constitutive model of W/PMMA composite microcellular foam material. The mechanical properties of W/PMMA composite microcellular foams with W contents of 0–60 wt % and relative densities of 0.15–0.55 were predicted. This work innovatively achieved the prediction of the mechanical properties of W/PMMA composite microcellular foams with different relative densities and W contents through extensive data fitting and model parameter identification, which can significantly reduce the number of experimental tests and increase the speed of analysis and prediction. We provided a basic method for optimizing the design of various W/PMMA composite microcellular foams. The calculation method in this paper can be further extended to other polymer systems and the addition of composite microcellular foams.

## 2. Materials and Methods

### 2.1. Preparation of W/PMMA Composite Microcellular Foams

The W particles and PMMA were fed into the melt-mixer (Torque Rheometer XSS-300, Shanghai Kechuang Rubber Plastics Machinery Set Ltd. Co., Shanghai, China) to prepare the composites from W particles and PMMA with various components. The melt-mixing was performed at 240 °C for 20 min. The rotor rotating speed was set to 60 r/min. Then, the composites were hot pressed at 170 °C for 60 min, thereby forming sheets with thicknesses of approximately 2 mm.

Then, the W/PMMA samples were placed into a specific mold with an inner height of 3 mm in a high-pressure autoclave under a fixed gas pressure of 18 MPa and at four foaming temperatures (65 °C, 80 °C, 95 °C and 120 °C). The physical blowing agent in this work was carbon dioxide. After complete saturation (approximately 12 h), cell growth was induced by the rapidly releasing the pressure within 1 s, and the foams were stabilized when the autoclave was cooled with a mixture of ice and water after 10 s of foaming.

### 2.2. Compressive Tests of W/PMMA Composite Microcellular Foams

The compressive modulus and strength of neat PMMA and W/PMMA composite microcellular foams were measured by a universal testing machine (QJ210A-5000N, Shanghai Instrument Co., Ltd. tilting technology, Shanghai, China) with a constant displacement rate of 0.5 mm/min under atmospheric pressure and ambient temperature. Each sample was tested three times, and the average value was calculated.

### 2.3. Constitutive Models of Foam Material

The compressive stress-strain curve of foam material has three distinct stages: linear elastic region, plateau region and densified region. Rusch [25] first presented one widely accepted model that can describe the relationship between compressive stress and strain: (1)$$\sigma \left(\varepsilon \right)=A\varepsilon^{m}+B\varepsilon^{n}\;\mathrm{with}\;0<m<1,\;1<n<\infty$$
where A, B, m, and n are curve-fitting constants that can be empirically determined. The first power formulation is used to fit the elastic-plateau region, while the second is used to model the densification region. Although this model represents the load-compression behavior of a flexible foam, it is inaccurate in describing the densification stage.

Then Gibson and Ashby proposed three equations to describe each region as follows [26]:(2)$$\sigma \left(\varepsilon \right)=E\varepsilon \;\mathrm{when}\;\sigma \leq \sigma_{\gamma }$$
(3)$$\sigma \left(\varepsilon \right)=\sigma_{\gamma }\;\mathrm{when}\;\varepsilon_{\gamma }\leq \varepsilon \leq \varepsilon_{D}\left(1-D^{-1}\right)+\varepsilon_{\gamma }$$
(4)$$\sigma \left(\varepsilon \right)=\sigma_{\gamma }D^{-1}{\left(\frac{\varepsilon_{D}}{\varepsilon_{D}-\varepsilon }\right)}^{m}\;\mathrm{when}\;\varepsilon >\varepsilon_{D}\left(1-D^{-1}\right)+\varepsilon_{\gamma }$$
where *E* is the elastic modulus, $\sigma_{\gamma }$ is the yield stress, $\varepsilon_{D}$ is the densification strain, and *D* and *m* are constants. However, the plateau region of this model has a constant value, and the stress-strain curve is not smooth at the boundaries of the two regions.

Liu and Subhash [28] proposed a mathematical formula to describe the stress-strain relationship of polyurethane foam in terms of the linear elastic, yielding plateau and densification stage of the foam under compressive loading:(5)$$\sigma \left(\varepsilon \right)=A\left(\frac{e^{\alpha \varepsilon }-1}{B+e^{\beta \varepsilon }}\right)+e^{C}\left(e^{\gamma \varepsilon }-1\right)$$
where parameters A, B, α and β are constant for a given initial density and strain rate and parameters *C* and *γ* depict the densification stage. Avalle et al. [29] used the exponential function, which provided a relevant improvement in the fit of the curve knee at the connection of the elastic region and the plateau region:(6)$$\sigma \left(\varepsilon \right)=A\left(1-e^{-\left(E/A\right)\varepsilon {\left(1-\varepsilon \right)}^{m}}\right)+B{\left(\frac{\varepsilon }{1-\varepsilon }\right)}^{n}$$
where: *A*, *B*, *E*, *m* and *n* are the parameters to be identified; note that the first item of the constitutive model characterizes the linear elasticity and yielding plateau region of the polyurethane foam during the compression process, and the second term represents the densification region.

Then, Voga [30] proposed a new phenomenological model named the Maxwell model to describe the compressive performance curve of foams. The stiffness coefficient *k* is equivalent to the elastic modulus of the foam, the damping coefficient *c* is equivalent to the plateau stress, which is the compressive strength, the stiffness *k_P_* represents the slope of the plateau region, and the stiffness coefficient *k_D_* is the 2-parameters (γ and n) exponential function of strain, which describes the densification region:(7)$$k_{D}\left(\varepsilon \right)=\gamma {\left(1-e^{\varepsilon }\right)}^{n}$$
(8)$$\sigma \left(\varepsilon \right)=e^{-\frac{k\varepsilon }{c}}\left(-1+e^{\frac{k\varepsilon }{c}}\right)c+\left[k_{P}+\gamma {\left(1-e^{\varepsilon }\right)}^{n}\right]\varepsilon$$

The values of parameter *n* could be only positive even numbers.

## 3. Results

### 3.1. Characterization of W/PMMA Composite Microcellular Foams

Figure 1 shows fracture micrographs of W/PMMA composites with various W contents. As the W content increased, the fracture surface of the composite material became rougher, and the W particle agglomeration became more substantial. Figure 2 shows a comparison of the theoretical density and experimental density of the W/PMMA composites. The degree of densification of the composites was greater than 97.5 %, and when the W content reached 73 wt % (15 vol %), the experimental density was significantly less than the theoretical density. Some voids can be clearly seen in Figure 1f; hence, the composite material is not dense, which may result in CO_2_ enrichment in the nondensified voids during the supercritical foaming process. This phenomenon will cause the formation of large cells, resulting in uneven foaming. Therefore, subsequent experimental studies did not consider the content of W particles of 73 wt %.

Figure 3 and Figure 4 show the morphology and density of the W/PMMA composite microcellular foams with different contents of W particles. The cell sizes of the W/PMMA composite microcellular foams are significantly smaller than those of the pure PMMA foams, and the cell density of the former is significantly greater. According to the classical nucleation theory [31], cell nucleation needs to overcome a large nucleation energy barrier, and cells are difficult to nucleate in a pure PMMA matrix. The remaining gas in the PMMA matrix was insufficient to generate new cells and instead entered existing cells. As a result, the cell sizes increased rapidly. With the added W particles, the nucleation energy barrier was greatly reduced, and additional cells were formed. After the cells stabilized, the gas remaining in the PMMA matrix that was unable to generate cells was distributed into the already formed cells, so that the cell sizes were smaller than those obtained by pure PMMA. W particles can act as heterogeneous nucleation agents to reduce the energy barrier and lead to higher cell densities (~5 × 10^10^ cells/cm^3^) and smaller cell sizes (~1.9 μm).

### 3.2. Mechanical Properties of W/PMMA Composite Microcellular Foams

Figure 5 shows the typical compressive stress–strain curves of pure PMMA foams and 20 wt % W/PMMA composite microcellular foams. The stress-strain curves of the microcellular foams have three distinct stages: linear elastic region, plateau region and densified region. Compared with the pure PMMA foams, the W/PMMA composite microcellular foams obtained under the same foaming conditions have a shorter plateau region and exhibit significantly greater elastic modulus (269.1 % greater) and compressive strength (123.9 % greater).

Because the mechanical properties of the foams have a significant correspondence with the density, the relationship between the elastic modulus, the compressive strength obtained by the stress-strain curves and the relative density is shown in Figure 6. Even under the same relative density, the elastic modulus and compressive strength values of the W/PMMA composite microcellular foams are approximately 50 % and 40 % greater than those of the pure PMMA foams, respectively. The W particles provide a point-like dispersion enhancement to the PMMA matrix. In addition, under the same relative density, the mechanical strength of the polymer foam depends not only on the matrix material, but also on its cell structure [32]. Several authors imply that smaller cell size might improve the mechanical strength [33,34]. Previous studies have shown that the heterogeneous nucleation of W particles can reduce the cell size of pure PMMA foams to approximately 1/3 and significantly homogenize the cell distribution, which greatly improves the mechanical properties of composite microcellular foams.

### 3.3. Constitutive Model of W/PMMA Composite Microcellular Foams

In this study, we chose the Maxwell model to describe the compressive curves of W/PMMA composite microcellular foams. According to the experimental curves of the foams, the least squares fitting method was used to obtain five parameters in the constitutive equation. The calculated parameters are listed in Table 1. The experimental and empirical fit curves are shown in Figure 7.

Although Voga [30] considered the parameter *c* representing the plateau stress, in this paper, the compressive strength (i.e., the plateau stress) should be based on the plateau stress of the curve fitted by the Maxwell constitutive equation, not the value of the parameter *c*.

20 wt % W/PMMA composite microcellular foams with relative densities of 0.200, 0.242, 0.320 and 0.522 were analyzed. The Maxwell model curves are compared with the experimental results. Figure 7 clearly shows that under the static compression of foams with different relative densities, the correlation coefficients between the calculated curves and the experimental curves are greater than 0.97, which fully demonstrates the high accuracy of the Maxwell model used in this paper.

## 4. Discussion

### 4.1. Parameter Identification of W/PMMA Composite Microcellular Foams

Figure 8 shows the relationships between the parameters in the Maxwell model and the relative density of the W/PMMA composite microcellular foams except the parameter *n*. This parameter must be a positive and even number: it is 4 for high density foam ($\rho_{r}>0.5)$ and 2 for low density foam ($0.15\leq \rho_{r}\leq 0.5$). The analysis of each parameter in the Maxwell model of W/PMMA composite microcellular foams shows that the parameter *k* is linear with the 1/3 power of the relative density, and the parameters *c*, *k_p_*, and *γ* are parabolically related to the relative density, the 1/3 power of the relative density, and the square of the relative density, respectively. The variance in the fitting of each parameter of W/PMMA composite microcellular foams is greater than 0.9. The relationship between each parameter and relative density is listed in Table 2.

Table 2 shows that the parameters in the Maxwell model are related not only to the relative density but also to the W content. Taking the W content as an independent variable, the coefficient of the relationship between each parameter and the relative density is the dependent variable, and the relationships can be found in Figure 9. The variance in the fitting of each coefficient of each parameter is greater than 0.95.

By fitting and analyzing the parameters corresponding to each W content in the W/PMMA composite microcellular foams, the relationship between relative density, W content and various model parameters can be obtained, as shown in Figure 10. The Maxwell model of the composite microcellular foams with W contents of 0~60 wt % and relative densities of 0.15~0.55 can be predicted, and simulated stress-strain curves can be obtained.

### 4.2. Application and Verification of the Constitutive Model

Figure 11 compares the predicted curve of the 50 wt % W/PMMA composite microcellular foam with a relative density of 0.184 to the experimental curve. The calculated parameters of the 50 wt % W/PMMA composite microcellular foam with a relative of 0.184 are shown in Table 3. The correlation coefficient of the model results and the experimental results is greater than 0.98, which verified the accuracy of the Maxwell model and the correct fitting relationship of each parameter.

## 5. Conclusions

We prepared tungsten/polymethyl-methacrylate (W/PMMA) composite microcellular foams by melt mixing and supercritical carbon dioxide foaming methods and tested the static compression mechanical properties of these microcellular foams. Then, we used the Maxwell model to fit the static compressive stress-strain curves of the W/PMMA composite microcellular foams. The experimental data accurately reflected the typical behavior of these microcellular foams and were in agreement with the model simulation results, which verified the accuracy of the model.

Through a series of model parameter identifications of different relative densities and W contents, we innovatively obtained the relationship between relative density, W content and model parameters in the static mechanical constitutive model of W/PMMA composite microcellular foams. The mechanical properties of W/PMMA composite microcellular foams with W contents of 0–60 wt % and relative densities of 0.15–0.55 can be predicted.

This work, for the first time, achieved the prediction of the mechanical properties of W/PMMA composite microcellular foams with different relative densities and W contents by extensive data fitting and model parameter identification, which can significantly reduce the number of experimental tests and increase the speed of analysis and prediction. We provided basic data and basic methods for optimizing the design of various W/PMMA composite microcellular foams and also provided a method for investigating the mechanical properties of composite microcellular foams, which can be further extended to other polymer systems and particle additions.

## Figures and Tables

**Figure 1 polymers-11-01136-f001:**
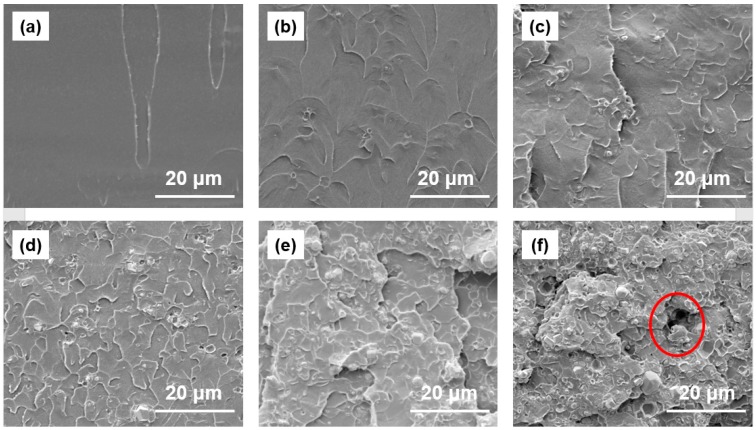
FESEM images of the fracture surfaces of (**a**) pure PMMA and (**b**) 10 wt % W/PMMA, (**c**) 20 wt % W/PMMA, (**d**) 40 wt % W/PMMA, (**e**) 60 wt % W/PMMA, and (**f**) 73 wt % W/PMMA composites.

**Figure 2 polymers-11-01136-f002:**
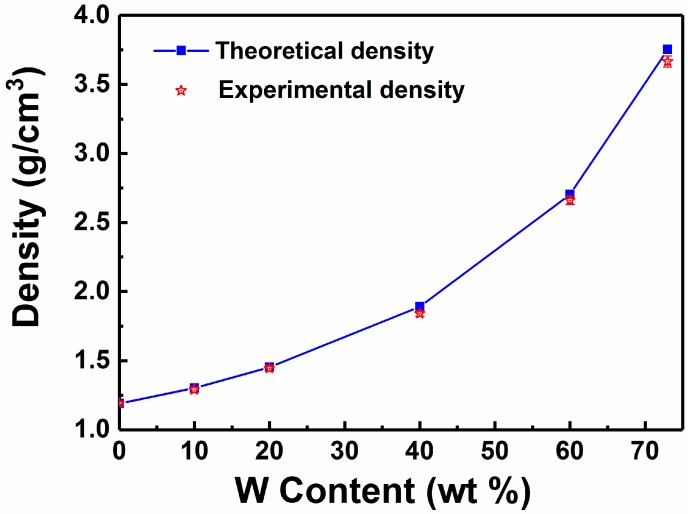
Comparison of theoretical density and experimental density of W/PMMA composites.

**Figure 3 polymers-11-01136-f003:**
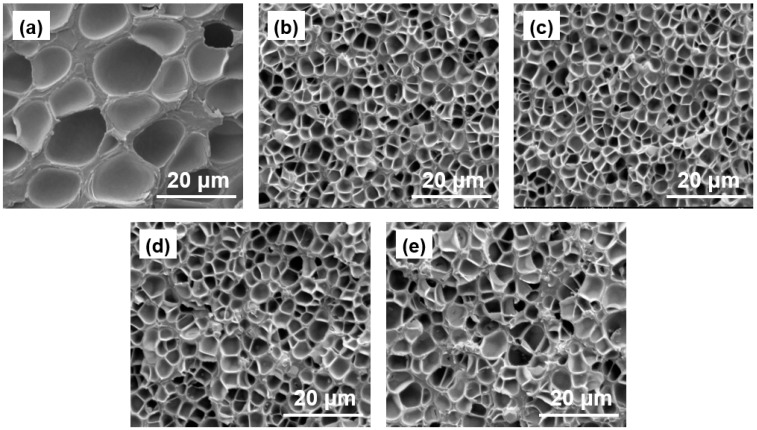
FESEM images of the typical cell morphology of W/PMMA composite microcellular foams with different contents of W particles produced at a saturation pressure of 18 MPa, a foaming temperature of 80 °C and a foaming time of 10 s: (**a**) pure PMMA foam, (**b**) 10 wt % W/PMMA foam, (**c**) 20 wt % W/PMMA foam, and (**d**) 40 wt % W/PMMA foam (**e**) 60 wt % W/PMMA foam.

**Figure 4 polymers-11-01136-f004:**
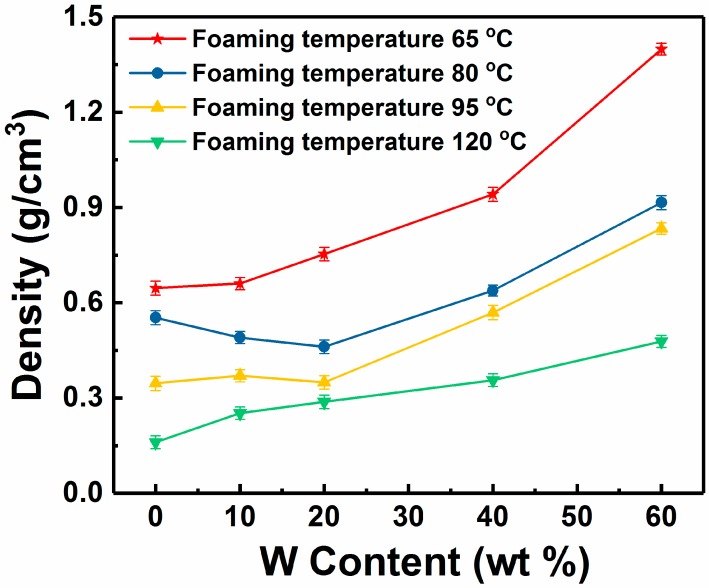
Density of W/PMMA composite microcellular foams with different contents of W particles.

**Figure 5 polymers-11-01136-f005:**
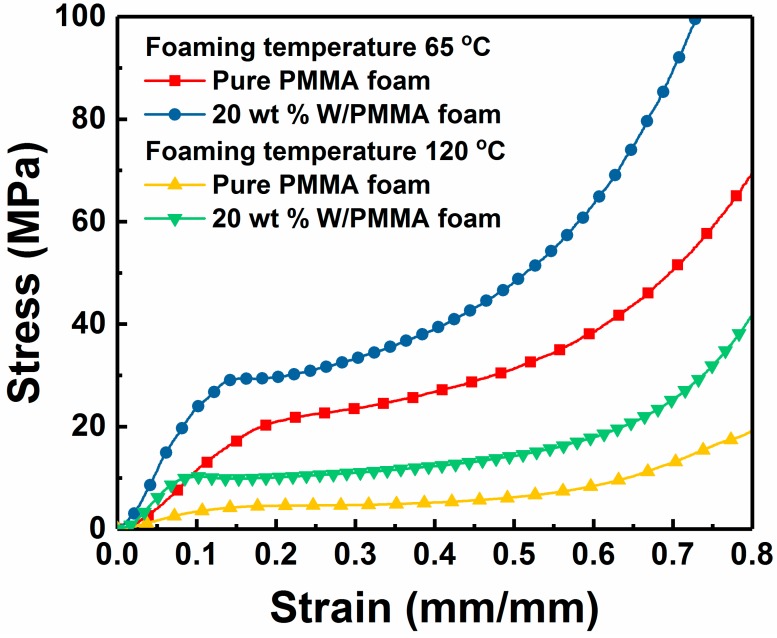
Compressive stress-strain curves of pure PMMA foams and W/PMMA composite microcellular foams containing 20 wt % W particles produced at a saturation pressure of 18 MPa, a foaming time of 10 s and foaming temperatures of 65 °C and 120 °C.

**Figure 6 polymers-11-01136-f006:**
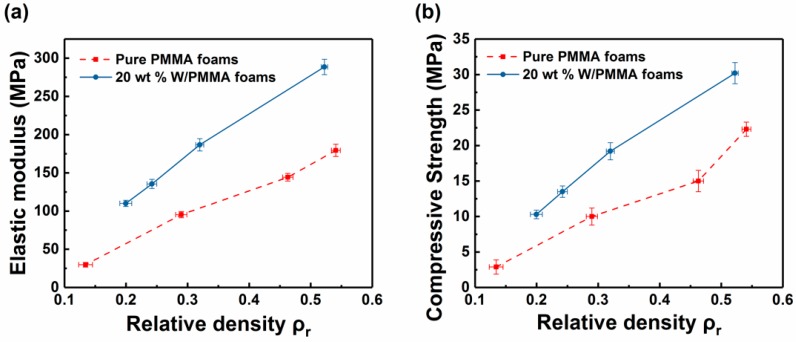
(**a**) Elastic modulus and (**b**) compressive strength of pure PMMA foams and 20 wt % W/PMMA composite microcellular foams with varying relative densities.

**Figure 7 polymers-11-01136-f007:**
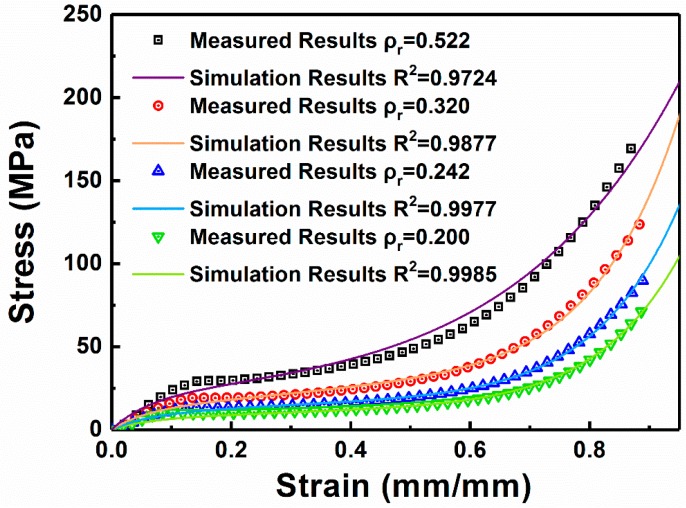
Comparison of experimental and empirical fit curves of 20 wt % W/PMMA composite microcellular foams at different relative densities.

**Figure 8 polymers-11-01136-f008:**
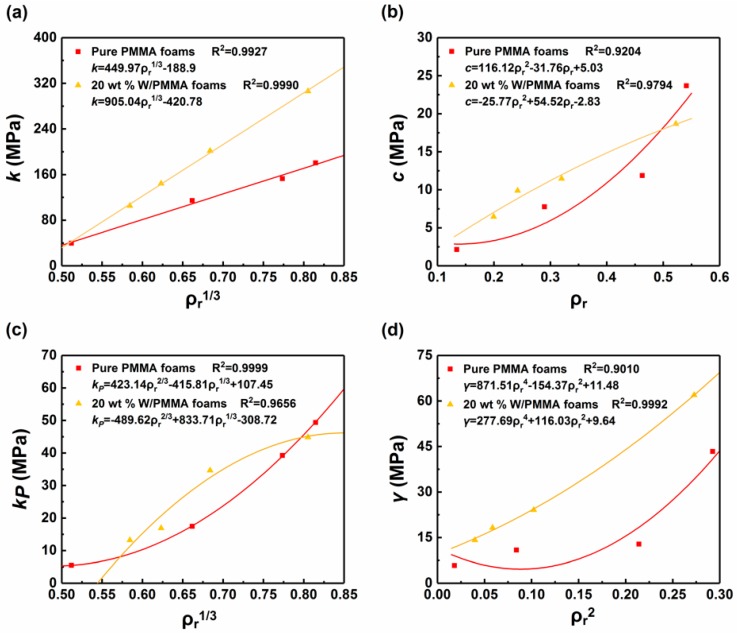
Relationships between the parameters in the Maxwell model and the relative density of the W/PMMA composite microcellular foams: (**a**) $k\sim \rho_{r}^{1/3}$; (**b**) $c\sim \rho_{r}$; (**c**) $k_{P}\sim \rho_{r}^{1/3}$, and (**d**) $\gamma \sim \rho_{r}^{2}$.

**Figure 9 polymers-11-01136-f009:**
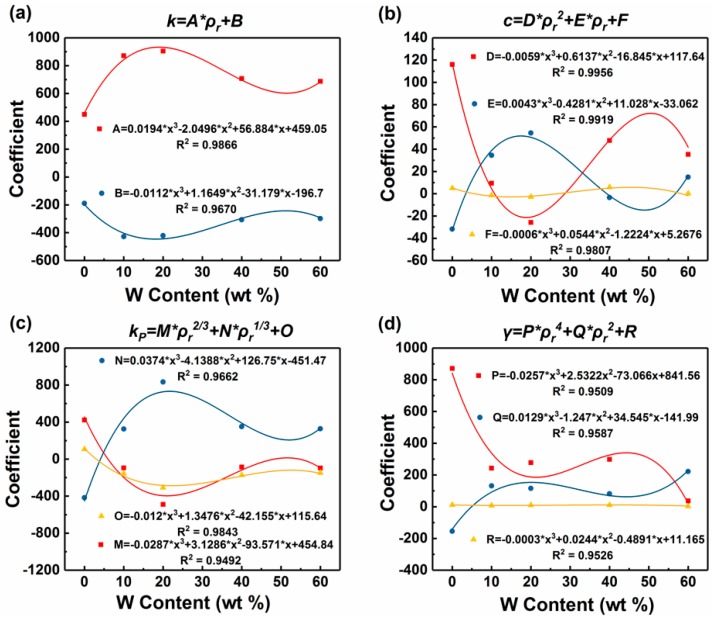
Relationship between the coefficient and W content (x) of each parameter in the Maxwell model: (**a**) coefficient *A*, *B* of *k*~*x*; (**b**) coefficient *D, E, F* of *c*~*x*; (**c**) coefficient *M*, *N*, *O* of $k_{P}$~*x*, and (**d**) coefficient *P*, *Q*, *R* of γ~*x*.

**Figure 10 polymers-11-01136-f010:**
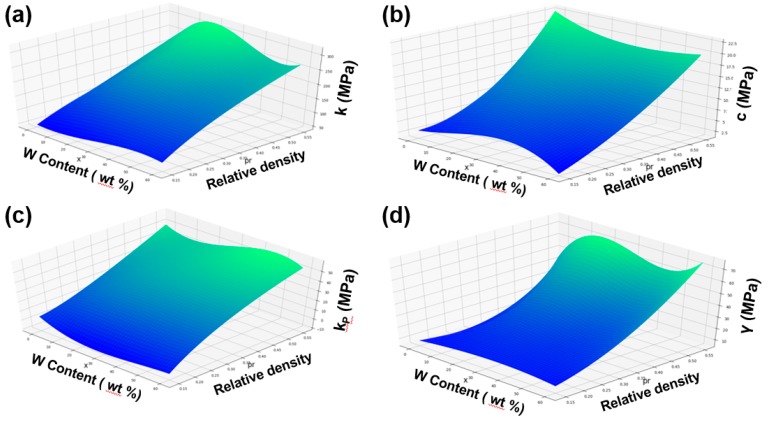
Parameters (**a**) k, (**b**) c, (**c**) $k_{P}$, and (**d**) γ in the Maxwell model as functions of relative density and W content.

**Figure 11 polymers-11-01136-f011:**
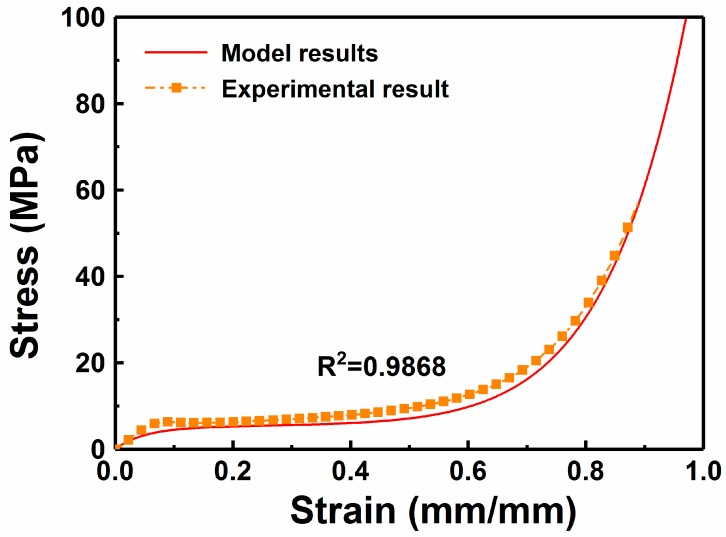
Comparison of experimental and simulated curves of the 50 wt % W/PMMA composite microcellular foam with a relative density *ρ_r_* = 0.184.

**Table 1 polymers-11-01136-t001:** Parameters in Maxwell model.

W Content in the Microcellular Foams	Relative Density (*ρ_r_*)	*k* (MPa)	*c* (MPa)	*k_p_* (MPa)	*n*	*γ* (MPa)
0 wt %	0.134	39.6	2.15	5.5	4	5.84
	0.290	114.3	7.77	17.4	4	10.9
	0.463	152.8	11.87	39.2	4	12.8
	0.541	180.5	23.68	49.4	2	43.4
10 wt %	0.196	78.4	5.14	0.7	4	10.8
	0.287	142.0	11.14	13.7	4	24.1
	0.381	211.4	11.68	31.5	4	28.9
	0.513	265.9	19.23	41.7	2	59.2
20 wt %	0.200	105.4	6.45	13.2	4	14.2
	0.242	144.2	9.88	16.9	4	18.2
	0.320	201.7	11.47	34.6	4	24.1
	0.522	306.3	18.66	44.8	2	62.0
40 wt %	0.194	103.6	6.96	3.7	4	14.0
	0.310	168.3	9.93	16.1	4	21.4
	0.347	191.9	10.02	42.7	4	24.6
	0.512	259.6	16.74	52.8	2	52.5
60 wt %	0.180	95.3	4.10	6.9	4	11.2
	0.314	148.4	8.40	21.8	4	20.1
	0.344	192.2	9.64	43.1	4	34.3
	0.526	260.0	17.92	53.3	2	67.0

**Table 2 polymers-11-01136-t002:** Relationship between each parameter and relative density in the Maxwell model.

W Content in the W/PMMA Microcellular Foams	$k=A*\rho_{r}+B$		$c=D*\rho_{r}^{2}+E*\rho_{r}+F$			$k_{P}=M*\rho_{r}^{2/3}+N*\rho_{r}^{1/3}+O$			$\gamma =P*\rho_{r}^{4}+Q*\rho_{r}^{2}+R$		
	A	B	D	E	F	M	N	O	P	Q	R
0 wt %	450	−189	116	−31.8	5.03	423	−416	107	871	−154	11.5
10 wt %	871	−428	9.48	34.5	−1.35	−95.2	325	−157	243	131	7.40
20 wt %	905	−421	−25.8	54.5	−2.83	−490	834	−309	278	116	9.64
40 wt %	707	−307	47.8	−3.43	5.91	−85.5	351	−172	298	81.4	10.6
60 wt %	687	−299	35.4	14.9	0.260	−95.8	328	−149	37.2	222	3.09

**Table 3 polymers-11-01136-t003:** Parameters of the 50 wt % W/PMMA composite microcellular foam in the Maxwell model.

Relative Density (ρ_r_)	*k* (MPa)	*c* (MPa)	*k_p_* (MPa)	*n*	*γ* (MPa)
0.184	100.5	4.94	2.0	4	13.3

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
