# Peer review of "Investigation of the Constitutive Model of W/PMMA Composite Microcellular Foams"

_polymers, 2019, doi:10.3390/polym11071136_

Reviewer 1 Report

The manuscript by Zhu et al. deals with the preparation of polymeric and composite foams, and characterizes empirically their stress-strain curves. The article is interesting, and worth publishing in MDPI Polymers upon some minor corrections. I have listed my comments and recommendations in chronological order.

Section 2.3: Please make sure to provide a description to all the parameters, after each equation (few of them are missing). I believe you meant to implement kD(ε) in equation 7, but you did not (either remove it from text or implement it).

Section 3.2, Ln.187: Please rephrase the following sentence to soften the claim:

“The smaller the cell size is, the better the mechanical strength. [27,28]”.

to

“Several authors imply that smaller cell size might improve the mechanical strength [27, 28]”.”

Although the cited  articles explicitly claim “Smaller bubbles are better for mechanical strength”, the experiments made are not exclusive to a bubble size change only, i.e. the effects are coming from contributions the authors neglected.

The cited article by Saha et al. 2005 [28] claims the use of different cross-linking of the polymeric foams used, which might result in differences of the mechanical strength, rather than the bubble size:

“H130 foam is highly cross-linked and HD130 foam is linear PVC foam. Special blowing agents and additives are added to produce H grade foam to have different microstructure as compared to HD grade while keeping both the foams of the same density. The additives basically control the degree of cross-linking during the polymerization phase.”

Chen et al. 2011 [27] uses composite materials with different thickness carbon nanotubes to compare the effect of the bubble size. Even the authors of the publication write:

“However, in the current study, the differences in the Young’s modulus and compressive strength of C100 and C20 can only be due to the aspect ratio of MWNTs used.

Therefore, the claim should be softened until further validation of the role of the bubble size, independently of other factors.

Section 3.2, Ln.188-191: Please add a citation(s) for the effect of foam polydispersity on mechanical strength.

Ln. 199: Please replace “simulated curves” with “empirical fit curves”.

Table 1: The parameter “c” given there is claimed to be “the damping coefficient c is equivalent to the plateau stress”. In the meantime, the parameters given in Table 1 for “0 wt%” are different from those given in Figure 6b and in Figure 5. Is there a technical mistake?

Also, how do you determine “plateau stress” in material with relative density 0.541 – the correlation coefficient seems OK, but the actual description of the curve appears to be poor. Maybe you should consider that the model does not work well below close packing of the bubbles, e.g. ρr > 0.35-0.40? I think this is also confirmed by the unexpected parabolic dependence of some of the parameters in Figure 8. If you remove the last point from all of the Figures b-d, you will get linear changes of the parameters? Is there an improvement of Figure 9, if you consider the bubble close packing as a requirement for the Maxwell behavior?

I have one additional question – what happens with the relative mechanical strength vs relative density of the composites, if you consider the mechanical strength of the tungsten and the PMMA?

Reviewer 2 Report

This paper investigated a constitutive model for tungsten/polymethyl-methacrylate (W/PMMA) composite microcellular foams prepared by melt mixing and supercritical  carbon dioxide foaming. The manuscript is interesting and the overall structure of the paper is good. However, the following improvements are recommended:

-The abbreviations in the Conclusions section should be explained at their first appearance. Conclusions must be readable independently without searching into the manuscript what certain abbreviations mean. In addition, the conclusions section should be written more concise (too many details are presented).

-The experimental density in Figure 2 should present the error bars. It is unlikely that all the measured samples have the same density.

-In this paper the authors discuss about Composite Microcellular Foams and their physical/mechanical properties. Accordingly, a brief introduction of composite polyurethane foam with different reinforced materials (glass fibers, aluminum microfibers, microspheres, biobased materials etc.) would add value to the present work (see Marsavina et al, Linul et al., Andersons et al. etc.). Please consider this.

- The standard after which the tests were performed should be presented.

- English language and style are fine/minor spell check required.

Summarizing, I recommend the paper for publication with Minor Revision.
